# PRETIE-Q Spanish Version and Affective-Functional Responses to Age-Based Strength Training in Older Women: An Exploratory Study

**DOI:** 10.3390/healthcare13233000

**Published:** 2025-11-21

**Authors:** Emilio Jofré-Saldía, Raúl Ricardo Festa, Álvaro Huerta Ojeda, Alejandro A. Candia, Claudio Farias-Valenzuela, Frano Giakoni-Ramírez, Victoria Torres Galaz, Sebastián Jannas-Vela, Denisse Valladares-Ide

**Affiliations:** 1Escuela de Ciencias de la Actividad Física, Facultad de Ciencias de la Rehabilitación y Calidad de Vida, Universidad San Sebastián, Santiago 8370040, Chile; 2Sports Performance Research, Rosario 2000, Argentina; rrfesta@gmail.com; 3Núcleo de Investigación en Salud, Actividad Física y Deporte ISAFYD, Universidad de Las Américas, Viña del Mar 2520000, Chile; ahuerta@udla.cl; 4Department for the Woman and Newborn Health Promotion, Universidad de Chile, Santiago 8320000, Chile; alejandrocandiah@uchile.cl; 5Escuela de Ciencias de la Actividad Física, el Deporte y la Salud, Universidad de Santiago de Chile (USACH), Santiago 9170022, Chile; claudio.farias.v@usach.cl; 6Facultad de Educación y Ciencias Sociales, Instituto del Deporte y Bienestar, Universidad Andres Bello, Santiago 7550000, Chile; frano.giakoni@unab.cl; 7Instituto de Ciencias de la Salud, Universidad de O’Higgins, Rancagua 2820000, Chile; victoria.torres@uoh.cl (V.T.G.); sebastian.jannas@uoh.cl (S.J.-V.); denisse.valladares@uoh.cl (D.V.-I.)

**Keywords:** PRETIE-Q, strength training, older women, functional performance, affective response

## Abstract

**Highlights:**

**What are the main findings?**
The PRETIE-Q was successfully adapted into Spanish. Tolerance increased during age-based strength training in older women, while Preference remained stable; neither was linked to affective valence or perceived exertion.Age-based strength training improved functional performance in older women.

**What is the implication of the main finding?**
Assessing Preference, Tolerance, affective valence, and perceived exertion in the context of strength training provides a comprehensive understanding of affective responses, which may promote adherence to training programs.Age-based strength training reduces health risks in older women, promoting greater independence.

**Abstract:**

Background: To promote exercise adherence, programs should consider not only functional effects but also a comprehensive affective response. This study analyzed the affective responses and functional performance in an age-based Block Strength Training (BST) for older women. Methods: Twenty-eight community-dwelling and Spanish-speaking older women participated in this study (age 68.39 ± 5.95 years) performed a 12-week age-based BST. Preference for and Tolerance of the Intensity of Exercise Questionnaire Spanish version (PRETIE-Q-Sv) was measured before and after each block four times. Additionally, Feeling Scale (FS) and Rating of Perceived Exertion session (sRPE) were measured. Functional performance was assessed before and after the BST using Timed Up and Go, Two-Minutes Step Test, and Five Times Stand-to-Sit Test. Statistical analyses included Cronbach’s alpha (*α*), Spearman’s (*ρ*, rho) and Pearson’s (*r*) coefficients, repeated-measures ANOVA or Friedman, and Paired comparison. The significance level was set at *p* < 0.05. Results: The PRETIE-Q-Sv showed acceptable internal consistency for both Preference (*α* = 0.80) and Tolerance (*α* = 0.78) constructs, with most items showing reliability with their construct scores (*ρ* ≥ 0.50). Preference remained stable (F (2.39, 64.59) = 2.64, *p* = 0.070, *η*^2^*_p_* = 0.09), and Tolerance increased (F (2.09, 56.35) = 11.84, *p* < 0.001, *η*^2^*_p_* = 0.30)., with overall average scores close to 22. These were not related to FS or sRPE (*ρ/r* = −0.02 to 0.14). All functional performance tests showed significant improvement after the BST (*p* < 0.001, *d/r_b_* > 0.80). Conclusions: The PRETIE-Q-Sv adapted well to the language. Preference and Tolerance remained at intermediate levels, with BST programming aligned to individual tendencies and showing no association with FS or sRPE, whereas the increase in Tolerance suggests that BST enhances older women’s ability to persist under uncomfortable effort. Improvements in functional performance support the effectiveness of the BST as a precision exercise dose.

## 1. Introduction

Adherence to exercise is essential in older adults. Programs should therefore address not only functional performance but also affective responses beyond pleasure and displeasure. According to ACMS’s Guidelines for Exercise Testing and Prescription, *Progression of physical activity should be individualized and tailored to tolerance and preference* (particularly in older adults) [1]. One tool to assess these dimensions is the Preference for and Tolerance of the Intensity of Exercise Questionnaire (PRETIE-Q), composed of 16 items (8 for Preference and 8 for Tolerance) on a 5-point response scale [2]. Preference refers to the predisposition to select a particular exercise intensity, usually in self-selected aerobic contexts [2,3], while Tolerance reflects the ability to persist at an imposed intensity despite discomfort or displeasure [2,4], making these constructs relevant to adherence and dropout prevention [3]. However, although the PRETIE-Q is available in several languages [2,5,6,7], to our knowledge, no Spanish version exists.

Recent proposals highlight the need for more specific programming in the WHO physical activity guidelines, particularly for strength training [8]. Although adherence in older adults is multifactorial [9], affective responses play a key role in adherence and motivation [10]. Strength training is increasingly recognized for its benefits in aging, and affective attitude is a predictor of participation [11]. However, affective responses research has focused mainly on aerobic exercise [12,13]. In strength training, higher affective valence and lower effort and arousal are reported with submaximal loads for a given number of repetitions compared to maximal loads [14], whereas training to failure increases perceived effort and discomfort even at the same load [15] due to mechanical and metabolic fatigue as repetitions progress [16]. Accordingly, current recommendations advise avoiding failure in older adults to reduce stress [17], thus avoiding a high level of effort and potentially favoring positive affective responses. However, although the affective response to resistance exercise has been studied [14,15,18], little is known about personal considerations such Preference and Tolerance assessed with PRETIE-Q in strength training, and no data exist on adults’ groups with an average age above 65, particularly older women [19].

Just as athletes require training specificity to meet the particular demands of their sport [20], emerging evidence suggests that a specific Block Strength Training (BST) based on age-related functional consequences and level of effort (not to failure) is effective in improving autonomy, quality of life, and functional performance in older women [21,22], thereby addressing the specific needs associated with aging; however, these studies have not focused on the affective response. Considering the low participation of older adults in sufficient strength training guidelines [23] and given women’s higher prevalence of frailty [24], understanding these aspects is essential to promote adherence and potentially reduce dropout. In this sense, recent evidence indicates that adjusting strength training according to individuals’ Preference and Tolerance enhances affective experience and is associated with higher session attendance among young adults [25]. Although our previous studies have relevant improvements following BST in older women [21,22], there remains a lack of evidence regarding the affective responses to such program and how these may influence individual considerations. In particular, information on Preference and Tolerance among older adults is scarce, particularly in women, and this limitation is further accentuated in Spanish-speaking populations due to the absence of a validated version of the corresponding assessment instrument.

This study aimed to (a) translate and validate the PRETIE-Q into Spanish and (b) examine affective responses during BST in older women while confirming its effectiveness in enhancing functional performance. It was hypothesized that the PRETIE-Q would demonstrate adequate adaptation to Spanish and that BST would elicit high affective valence, low-to-moderate perceived effort, stable or increased Preference and Tolerance, and improvements in functional performance.

## 2. Materials and Methods

### 2.1. Participants

A total of twenty-eight community-dwelling older women, all Spanish-speaking and residing in the Metropolitan Region of Chile, were recruited to participate in this study (age, 68.39 ± 5.95 years; body mass, 70.55 ± 14.82 kg; height, 153.39 ± 5.05 cm; body mass index, 29.88 ± 5.42 kg/m^2^). The recruitment of older women was carried out through advertising and informational meetings. The group was considered relatively homogeneous. According to conventional coefficients of variation thresholds, age (8.69%) and height (3.29%) showed low variability, whereas body mass (21.01%) and body mass index (18.17%) showed moderate variability. Overall, the coefficients of variation values indicated an acceptable level of group homogeneity. The following inclusion criteria were applied for participation in this study: (i) age between 60 and 80 years; (ii) non-smoker; (iii) no exercise program during the last 6 months before this study; (iv) evaluation and approval of the medical examination. In addition, the following exclusion criteria were applied: (i) inability to move; (ii) illness; (iii) disease; (iv) abnormal electrocardiogram. All participants provided written informed consent. This study was approved by the Ethical-Scientific Committee of Santiago de Chile University, Santiago, Chile (Ethics Approval No.: 570-2024). The present study was conducted in accordance with the ethical principles of the Declaration of Helsinki and the Ethical Standards for Exercise and Sport Science [26].

#### Sample Size

The sample size was calculated a priori using G*Power 3.1.9.6 (F test: ANOVA, repeated measures, within factors, compute required sample size) [27]. A medium effect size (f = 0.25), significance level (α = 0.05), and statistical power (1 − β = 0.90) were assumed. The calculation was based on a single-group design and the variable with the highest number of measurements (4 time points), assuming a correlation between repeated measures of *r* = 0.70 and a nonsphericity correction of ε = 0.70. The minimum required sample was 24 participants; however, a total of 28 participants were recruited to account for potential dropouts or missing data.

### 2.2. Study Design

The present study was designed as a longitudinal, quasi-experimental trial with repeated measures, and was divided into two phases,

(1)Translation of the PRETIE-Q from its original and validated English version [2] to a Spanish version; and(2)Older women performing a BST based on age-related functional consequences and level of effort.

#### 2.2.1. Phase 1: Translation of the PRETIE-Q

The translation of the original PRETIE-Q from English to Spanish was carried out through a four-step process (Figure 1; upper panel):(1)Two Chilean English-Spanish translators performed an independent preliminary translation from English to Spanish of the original questionnaire [2].(2)A third Chilean English-Spanish translator consolidated both independent preliminary translations from English to Spanish of the original questionnaire [2].(3)A Chilean PhD in Sports Sciences with experience in researching psychometric variables reviewed the PRETIE-Q-Sv.(4)For the generation of the final translation of PRETIE-Q-Sv, disagreements were resolved by consensus between Chilean translators and the PhD in sports sciences.

In summary, the questionnaire comprises 16 items, with even-numbered items assessing exercise intensity preference and odd-numbered items assessing tolerance to intense exercise. Specifically, items 6, 10, 14, and 16 assess a preference for high-intensity exercise, whereas items 2, 4, 8, and 12 assess a preference for low-intensity exercise. Conversely, items 5, 7, 11, and 15 indicate higher tolerance for intense exercise, while items 1, 3, 9, and 13 indicate lower tolerance. Minor adjustments to specific item terms were agreed upon by the translators and the sports science expert to achieve greater cultural and linguistic adaptation to Spanish and to the general population. For example, item 3 refers to “breathing very hard,” which literally translates as “respirando muy fuerte.” To avoid confusion with a voluntary action, this expression was rendered as “me cuesta mucho respirar”. In item 5, the term “exhausted” was translated as “agotado(a)”*,* a more colloquial and context-appropriate expression. Likewise, the phrase “I usually ease off some” in item 13 was translated as “bajo un poco la intensidad”. Finally, in item 14, the word “harder,” which could be interpreted as “intenso” or “duro”, was translated as “exigente” to ensure a more accurate interpretation of the sentence. Further details can be found in the translated questionnaire provided in the Appendix A.

#### 2.2.2. Phase 2: Intervention

A familiarization process with a comprehensive reading of PRETIE-Q-Sv, functional performance tests, strength exercises, and the affective and perceptual scales to use was carried out. Subsequently, the BST was carried out over three 4-week blocks (12 weeks total), with two sessions per week on non-consecutive days, in groups of 10–14 participants. The blocks (block 1, B1; block 2, B2; and block 3, B3) were designed according to the model of functional consequences of age-related sarcopenia (i.e., ↓ strength; ↓ power; ↓ muscular endurance) proposed by Hunter et al. [28] and in line with previous studies [21,22]. The novelty with these studies and the present one was the inverted order of B2 and B3. A professional with experience in training older adults was responsible for leading the sessions (including a standardized 10 min warm-up and cool-down section, respectively) and completing the attendance register of the participants. The PRETIE-Q-Sv was applied before training program and after the last session of each block of the BST program (i.e., three times) in a comfortable environment, guaranteeing privacy and availability of personalized assistance from researcher. Additionally, affective valence using the Feeling Scale (FS) [29] and session Rating of Perceived Exertion (sRPE) [30] were reported after each session, and averaged to a single value per block. Finally, functional performance was also evaluated before and after the BST (Figure 1; middle-bottom panel).

### 2.3. BST Protocol

Traditional multi-joint exercises of upper and lower limbs typically reach between ~10 to 25 repetitions at moderate load (~60–75% of one repetition maximum; i.e., %1RM) [31,32], therefore the BST external loads were individualized using the number repetition maximum (nRM) test [33] bi- or unilaterally depending on the block (i.e., 10 RM or 15 RM) every 8 sessions; which were performed using a Smith machine and kettlebells (ILUS Fitness^®^, Santiago, Chile). The blocks were oriented towards force peak (neuromuscular coordination), force tolerance (muscular endurance), power peak (high neuromuscular activation), regulated by cadence (KDM-2 digital metronome; Korg Inc., Tokyo, Japan) and load according to current recommendations for older adults [34]. The total volumes of each session (i.e., sets × repetitions × exercises) were 90, 180, and 60, with an average intensity of 10 RM, 15 RM, and 10 RM for B1, B2, and B3, respectively. The low to high level of effort (i.e., not to failure reps) was regulated by 0 to 10 RPE scale (≤7 and 8–9; it refers immediately after set completion, being distinct from sRPE) and the percentage of maximum repetitions (%rep; 50–80%) which are linked to intra-set velocity loss [31,35,36]. In order to maintain the programmed effort adjustments to the external load made as needed. For more details, see the lower panel of Figure 1.

### 2.4. Measurements

#### 2.4.1. PRETIE-Q Spanish Version, Affective Valence and Perceived Exertion

The PRETIE-Q consists of two 8-item scales for Preference and Tolerance, respectively, which are accompanied by a 5-point response scale ranging *from “*I totally disagree*” to “*I totally agree*”* [2]. On the Preference scale, items 6, 10, 14, and 16 measure preference for high-intensity exercise, and items 2, 4, 8, and 12 measure preference for low-intensity exercise. On the Tolerance scale, items 5, 7, 11, and 15 measure high tolerance, and items 1, 3, 9, and 13 measure low tolerance for intense exercise. Items indicative of preference for low intensity (items 2, 4, 8, 12) and items indicative of low tolerance (items 1, 3, 9, 13) are reversed-scored [5,37]. The score range for each scale is 8–40 points. The PRETIE-Q used was the Spanish version translated in Phase 1 (Study Design) of the present study (see Appendix A). The PRETIE-Q-Sv was administered prior to the functional performance and B1 1RM tests to avoid potential bias in Preference and Tolerance scores. During the intervention, it was completed in the final session of each block to more accurately reflect the effects of each training phase (see Figure 1, bottom panel). The FS was used to assess affective valence to exercise [29], which is a scale that has an 11-point bipolar good/bad format, ranging from +5 to −5. Verbal anchors are provided at the 0 point, and at all odd integers +5 = very good, +3 = good, +1 = fairly good, 0 = neutral, −1 = fairly bad, −3 = bad, and −5 = very bad. The sRPE format designed by Foster et al. (scale from 0 or Rest to 10 or Maximal) was used to measure perceived exertion, where older women’s were asked to provide a rating of the overall difficulty of the entire session, not at a particular time, by asking them “how was your workout?” [30]. The qualitative categories of these instruments were translated using the same steps described in section study design for translation of the PRETIE-Q. Both FS and sRPE were evaluated approximately 30 min after the end of each training session [30], taking the average of the eight sessions of each block for subsequent analysis.

#### 2.4.2. Functional Performance Assessment

Functional performance was assessed through simple and practical tests [38]. The Timed Up and Go (TUG) test was administered to determine lower-limb functional capacity and dynamic mobility. This assessment records the time (Polar^®^ Electro Oy V800, Kempele, Finland) required to stand up, walk 3 m, turn around an obstacle, return to the chair, and sit down. A performance time greater than 12 s has been recognized as a clinically meaningful threshold, reflecting an increased risk of fractures in older women [39]. The Two-Minute Step Test (2MST) was applied to assess endurance performance. This test records the number of times the right knee is raised to the level between the patella and the anterior superior iliac spine. An outcome of 60 steps or less denotes functional impairment among older individuals [40]. The Five Times Sit-to-Stand Test (5-STS) was used for estimating lower body strength. This test measures the time (Polar^®^ V800, Finland) needed to complete five repetitions of rising from and sitting down on an armless chair as quickly as possible. Taking more than 11.64 s to complete the test may be indicative of lower muscle strength in independent older women [41].

### 2.5. Statistical Analysis

Data are presented as mean (M) and standard deviation (SD), unless otherwise indicated. When non-parametric analyses were required, data are presented as median and Interquartile Range (IQR). Normality was assessed using the Shapiro–Wilk test (n < 50) or the Kolmogorov–Smirnov test (n ≥ 50), with *p* > 0.05 indicating normal distribution. For repeated-measures analyses, the assumption of sphericity was verified using Mauchly’s test, applying the Greenhouse–Geisser correction when violated. The assumption of homogeneity of variances (Levene’s test) was not applicable in this study, as all participants (n = 28) underwent the same intervention and no between-subject factor or control group was included. The internal consistency of the Preference and Tolerance scales was evaluated using Cronbach’s alpha coefficient (*α*). The relationships between Preference and Tolerance scores and the items of each scale were examined using Pearson’s correlation coefficient (*r*) when both variables were normally distributed, or Spearman’s rank correlation coefficient (*ρ*, rho) when the normality assumption was violated [42]. Both consistency and reliability were evaluated in a single sample (n = 112; four observations per subject).

Depending on the underlying data distribution and the extent of the violation of normality [43], either a repeated-measures ANOVA (parametric) or a Friedman test (non-parametric) was used to analyze the effects of training blocks on Preference and Tolerance scores (pre- and during BST), as well as on FS and sRPE (during BST). Post hoc analyses were conducted with Tukey’s test (for ANOVA) or Durbin–Conover pairwise comparisons (for Friedman). Accordingly, ANOVA results reflect mean differences, whereas Friedman and Wilcoxon tests reflect differences in median ranks. This mixed approach was chosen to maximize statistical power while respecting the assumptions of each variable.

In addition, *r* or *ρ* were used to examine the relationship between Preference and Tolerance scores with FS and sRPE during BST (n = 84; three observations per subject). After confirming the normality of the difference scores for each pair [44], either a paired-samples *t*-test or a Wilcoxon rank test was conducted to evaluate the effects of the BST on functional performance, as measured by the TUG, 2MST, and 5-STS.

The magnitude of correlation coefficients was interpreted by convention as follows: *r/ρ* < 0.30 (small), ≥0.30 and <0.50 (medium), and ≥0.50 (large) [45]. Effect sizes for parametric tests were estimated using partial eta squared (*η*^2^*_p_*) [0.01 (small), 0.06 (medium) and 0.14 (large)], and Cohen’s *d* [0.20 (small), 0.50 (medium), and 0.80 (large)] [45,46]. For non-parametric ANOVA analyses (Friedman tests), no standard effect size was calculated. For non-parametric *t*-tests (Wilcoxon rank test), the rank-biserial correlation (*r_b_*) was reported and interpreted using Cohen’s conventions for *r* [0.10 (small), 0.30 (medium), 0.50 (large)] [45]. Statistical significance was set at *p* < 0.05.

All statistical analyses were performed using Jamovi for macOS (The jamovi project (2024). *jamovi* (Version 2.5) [Computer Software]. Retrieved from https://www.jamovi.org; accessed on: 24 March 2024). Figures were generated using Microsoft PowerPoint 2016 (Microsoft Corporation, Redmond, WA, USA) and GraphPad Prism (version 8.0.1; GraphPad Software, San Diego, CA, USA). 

## 3. Results

### 3.1. Translation of the PRETIE-Q

The final consolidated version of PRETIE-Q-Sv can be seen in the Appendix A. The internal consistency for the Preference and Tolerance items was *α* 0.80 (n = 112) and 0.78 (n = 112), respectively. The normality of the PRETIE-Q-Sv total scores was evaluated using the Kolmogorov–Smirnov test, revealing normal distributions for both Preference and Tolerance constructs (*p* > 0.05). In contrast, all individual items showed non-normal distributions (*p* < 0.01). Consequently, non-parametric rank correlation (*ρ*) was used to examine the relationships between items and their corresponding total scores. No item had a negative contribution to internal consistency. Most of the individual questions showed significant medium to large positive correlations (*ρ* = 0.35–0.71, *p* < 0.001) with the scores of their respective scales, except for item 9, which showed a smaller but still significant correlation (*ρ* = 0.24, *p* < 0.05) (Figure 2).

### 3.2. Intervention

Older women attended more than 80% of all BST sessions on average, with several participants achieving attendance rates above 90%. Since the absences were due to reasons unrelated to the program, these results indicate a high level of adherence to the strength training program.

#### 3.2.1. PRETIE-Q Spanish Version, Affective Valence and Perceived Exertion

The overall scores for Preference and Tolerance were 22.12 ± 7.37 and 22.36 ± 6.37, respectively. According to the Shapiro–Wilk test (n = 28), both Preference and Tolerance scores met the assumption of normality (W > 0.92, *p* > 0.05) at all four time points (Pre, B1, B2, and B3). Therefore, a repeated-measures ANOVA was applied to examine within-subject changes across the intervention.

For Preference, Mauchly’s test indicated a violation of sphericity (W = 0.53, *p* = 0.007); therefore, the Greenhouse–Geisser correction (ε = 0.80) was applied. The repeated-measures ANOVA revealed no statistically significant differences across time points (F (2.39, 64.59) = 2.64, *p* = 0.070, *η*^2^*_p_* = 0.09), although a medium-to-large effect size was observed. Mean scores ranged from 19.93 ± 5.41 (Pre) to 24.18 ± 8.63 (B3).

For Tolerance, Mauchly’s test also indicated violation of sphericity (W = 0.17, *p* < 0.001); thus, the Greenhouse–Geisser correction (ε = 0.70) was used. The repeated-measures ANOVA revealed a significant main effect of time (F (2.09, 56.35) = 11.84, *p* < 0.001, *η*^2^*_p_* = 0.30). Post hoc Tukey comparisons showed that Tolerance scores increased significantly from Pre to B2 (*p* = 0.005) and B3 (*p* = 0.001), and from B1 to B2 (*p* = 0.009) and B3 (*p* = 0.002). No significant differences were observed between Pre and B1 (*p* = 0.988) or B2 and B3 (*p* = 0.902). The results for both variables are reported in Table 1.

According to the Shapiro–Wilk test (n = 28), FS scores showed non-normal distributions at all time points (W < 0.69, *p* < 0.001). Therefore, the Friedman test was applied to analyze changes across training blocks. The analysis revealed a significant main effect of time (χ^2^(2) = 10.72, *p* = 0.005). Post hoc Durbin–Conover comparisons indicated that FS scores increased significantly from B1 to B3 (*p* < 0.001), while no significant differences were observed between B1 and B2 (*p* = 0.059) or between B2 and B3 (*p* = 0.107).

RPE scores showed a non-normal distribution at B3 (n = 28; W = 0.92, *p* = 0.043), whereas B1 and B2 were normally distributed (W > 0.94, *p* > 0.05). Therefore, changes across the intervention were analyzed using the Friedman test, which revealed a significant main effect of time (χ^2^(2) = 46.29, *p* < 0.001). Post hoc Durbin–Conover comparisons showed that RPE increased from B1 to B2 (*p* < 0.001), decreased from B2 to B3 (*p* < 0.001), and decreased from B1 to B3 (*p* < 0.001). The results for both variables are reported in Table 2.

Finally, the relationships among variables during BST were analyzed. The normality of Preference, Tolerance, FS, and sRPE scores was assessed using the Kolmogorov–Smirnov test (n = 84), revealing normal distributions for Preference, Tolerance, and sRPE (*p* > 0.05), but not for FS (*p* < 0.001). Consequently, parametric (*r*) or non-parametric (*ρ*) correlation coefficients were applied as appropriate. Preference did not significantly correlate with FS (*ρ* = 0.14) or sRPE (r = −0.02), and Tolerance scores likewise did not correlate with FS (*ρ* = −0.01) or sRPE (*r* = 0.05).

#### 3.2.2. Functional Performance

According to the Shapiro–Wilk test (n = 28), the distribution of the pre-to-post intervention differences for the TUG and 2MST performance measures were found to be non-normal (W < 0.88, *p* < 0.01). In contrast, 5-STS difference exhibited a normal distribution (W = 0.97, *p* = 0.527). Consequently, a Wilcoxon rank test and a paired-samples *t*-test were applied, respectively. Functional performance showed a significant improvement across all tests following the BST (Figure 3). Specifically, the TUG performance significantly improved from a baseline median of 6.86 s (IQR: 6.42–7.68) to 6.18 s (IQR: 5.61–6.90) post-intervention (*p* < 0.001, *r_b_* = 0.94). Similarly, the 2MST demonstrated significant improvement, increasing the median number of steps from 102.50 (IQR: 92.25–110.50) to 123.00 (IQR: 107.75–133.50) (*p* < 0.001, *r_b_* = −1.00). Finally, the 5-STS also showed a statistically significant improvement, with the mean time decreasing from 9.83 ± 3.11 s to 7.96 ± 3.22 s (*p* < 0.001, *d* = 1.25).

## 4. Discussion

The present study demonstrates that the PRETIE-Q-Sv was successfully adapted for language. Preference and Tolerance remained at intermediate levels throughout the intervention, reflecting the moderate external load and level of effort programmed in the BST. However, the increase in the Tolerance score suggests that BST promotes improvements in the ability to persist under uncomfortable effort. Affective valence (i.e., FS) remained consistently high despite minor fluctuations, whereas perceived exertion varied across blocks according to the level of effort applied (%rep); however, neither was associated with construct scores. Importantly, following BST, older women achieved significant improvements in functional performance, further moving beyond established risk cutoffs. Altogether, these findings support BST as a precision-dose person-centered strategy to promote exercise adherence in older women, reframing the traditional concept of “no pain, no gain” toward “no pain, optimized gain.”

### 4.1. Internal Consistency, Reliability and Scores of PRETIE-Q-Sv

The acceptable-to-good internal consistency of the PRETIE-Q-Sv Preference and Tolerance scales is similar to that of the original English version [2], as well as the Brazilian-Portuguese [5], European-Portuguese [6], and Chinese versions [7]. Regarding reliability between constructs and items, it is comparable to reported for the original English and the Brazilian-Portuguese version [2,5]. Although no items had a negative contribution in the present study as in the English version [2], items 9 and 13 showed the lowest correlations. This is similar to the findings for the Brazilian-Portuguese version, where item 7 had a negative contribution and a low correlation [5]. These items may require certain experience in a given activity, as proposed by other authors [47].

Previous studies have reported Preference and Tolerance scores, but our results are not directly comparable with those from the Chinese and abbreviated European-Portuguese versions, as these use a different total number of items than the original 16-item questionnaire [6,7]. Smirmaul et al. reported average Preference and Tolerance values of 25 to 30 using the Brazilian-Portuguese version in a mixed group of young adults [5]. However, the English version is the most widely applied. Ekkekakis et al. [3,37] reported average scores of 22–24 for Preference and 20–23 for Tolerance in sedentary middle-aged women and a large sample of female university students. More recently, Box & Petruzzello reported values near 27 for both constructs in a mixed group of young adults (82% exercisers), with women reporting slightly lower values than men [48]. Overall, average scores reported are between 20 and 30 in both constructs; in our study, repeated assessments in the BST context remained closer to 22, although individual variability must be considered (CV~30%). Little information about the PRETIE-Q in older adults [19] with only one study reporting values of 26 and 25 for Preference and Tolerance, respectively, in a physically active mixed group aged 64 years [49]. Our study is the first to report data in adults with an average age above 65. In general, Preference and Tolerance scores typically fall at moderate levels, reflecting a tendency toward moderate exercise intensity and a moderate capacity to cope with discomfort, which is consistent with the BST programming in this study. From a practical standpoint, the PRETIE-Q supports tailoring exercise programming to individual affective dispositions, as has been recently proposed [25].

### 4.2. Affective Response to BST

PRETIE-Q remains underexplored in strength training [19]. Tolerance scores have been positively related to blood lactate and pain perception, while Preference is related only to blood lactate after circuit resistance training in young adults [50]. Positive relationships have been found between Preference and Tolerance with general arousal, training arousal, and training volume, as well as between Preference and general FS, in strength training to failure [51]. The combined effect of arousal and FS suggests that greater Preference or Tolerance is associated with a better affective response [51]. Teixeira et al. highlighted that exercise instructions emphasizing self-regulation are a practical method to enhance affective experience and promote greater engagement [25]. However, the PRETIE-Q is typically administered as a single or pre-post assessment, and few studies have included more than two measurement points. For example, Flack et al. assessed PRETIE-Q pre-post and after a washout period of physical activity recommendations in sedentary subjects aged 18–49 years (83% female), showing that increase in Tolerance were linked to incentive sensitization of exercise reinforcement [52]. Our study is the first to assess Preference and Tolerance on more than two occasions during strength training in older women, showing stable Preference but increased Tolerance, suggesting that BST improves persistence under discomfort without requiring maximal effort. Internal load (i.e., sRPE) and affective valence (i.e., FS) were not related to PRETIE-Q-Sv constructs during the BST, potentially due to programming (e.g., not to failure), highlighting Preference and Tolerance as key by incorporating the individual into the exercise programming, in line with ACSM Guidelines [1]. As the BST program does not affect Preference but increases Tolerance while remaining at moderate levels, it appears to be a promising strategy to promote an overall positive affective response by addressing both motivational aspects and the ability to withstand effort through BST, in line with individual tendencies. Given the greater vulnerability of older women compared to men [24] and the generally low adherence of older adults to traditional strength programs [23], our findings suggest that BST represents a promising approach to enhance adherence by aligning with individual considerations for exercise programming in this population.

Regarding programming effects on affective responses, valence and perception of effort may remain higher and lower, respectively, when maximal repetitions are avoided [14]. Emanuel et al. showed FS decreases and related to high RPE depending on %rep performed [18], with similar effects when applied heavier loads [53]. Similarly, Varela-Olalla et al. reported exponential RPE increases as %rep rises [35]. The present study may show smaller alterations due to measurement timing, but our results align with findings indicating that sRPE effectively monitors different strength type sessions despite lower absolute values than post-set RPE [54]. High FS and low-to-moderate sRPE can be explained by applied effort, likely avoiding high mechanical and metabolic fatigue [16], thereby supporting high attendance and adherence rates (>80%) observed in the BST program. Finally, although implemented in a controlled setting, BST emphasizes programming over specialized equipment, and its exercises mirror those commonly used in community-based strength programs. This suggests that the observed effects are likely achievable in everyday practice, thereby enhancing this study’s ecological validity.

### 4.3. Functional Performance Response to BST

Strength training not performed to failure is emerging as effective for improving functional performance in older adults [55,56], particularly with BST [21,22]. Our study applied a model based on age-related sarcopenia consequences [28], which is important as periodized programs outperform non-periodized ones [57]. Functional performance average improvements after BST enabled older women to move away from cut-offs for fracture risk (37 to 46% Δ), functional impairment (67 to 102% Δ), and reduced muscle strength (16 to 32% Δ) [39,40,41], positively impacting health risks and confirming BST effectiveness. Our findings are consistent with those of Cadore et al. and Marques et al., who reported that non-failure training programs—using 50% of the total repetitions or a 20% intra-set velocity loss threshold—led to improvements in 5-STS and TUG performance in older adults [55,58]. Specifically, Jofré-Saldía et al. recently reported that a BST protocol similar to that applied in the present study produced significant improvements in handgrip strength, TUG, 2MST, 5-STS, and walking speed in a group of 40 older women compared with standard physical activity recommendations [22]. These findings further support the effectiveness of the BST program over general recommendations [8]. Emergent evidence indicates that outcomes are task-specific in strength training for older adults [59]; therefore, BST or most exercises applied here can be recommended to improve functional performance. Additionally, as women may require lower exercise doses than men to achieve similar benefits [60], future studies should compare BST effects in men and mixed samples.

### 4.4. Future Directions and Limitations

Although the internal consistency and reliability of the PRETIE-Q-Sv are promising, these results should be interpreted with caution, as the instrument was applied only in older women. Furthermore, we acknowledge the limitation posed by the absence of a control group and randomization; however, the PRETIE-Q-Sv assessment conducted prior to the BST intervention may serve as a form of internal control. Future research should extend its application to other populations, particularly young adults and men, and include a control group to allow comparison of the comprehensive affective responses to BST. Since PRETIE-Q was originally developed in the context of aerobic exercise [2,3,4] further studies should explore possible adjustments to enhance its specificity for strength training according to the level of effort. In addition, the increase observed in Tolerance during BST highlights the need to investigate whether such changes are associated with biomarkers related to mood and pain regulation.

## 5. Conclusions

The PRETIE-Q-Sv demonstrated successful linguistic adaptation and applicability in older women. Preference and Tolerance remained at moderate scores, with Tolerance increasing, reflecting BST programming and improved ability to persist under uncomfortable effort. These affective responses were independent from FS and sRPE, underscoring the importance of analyzing them as separate constructs when evaluating comprehensive affective responses. In parallel, older women achieved improvements in functional performance, reaffirming the effectiveness of BST and moving away from the limits of health risk.

## Figures and Tables

**Figure 1 healthcare-13-03000-f001:**
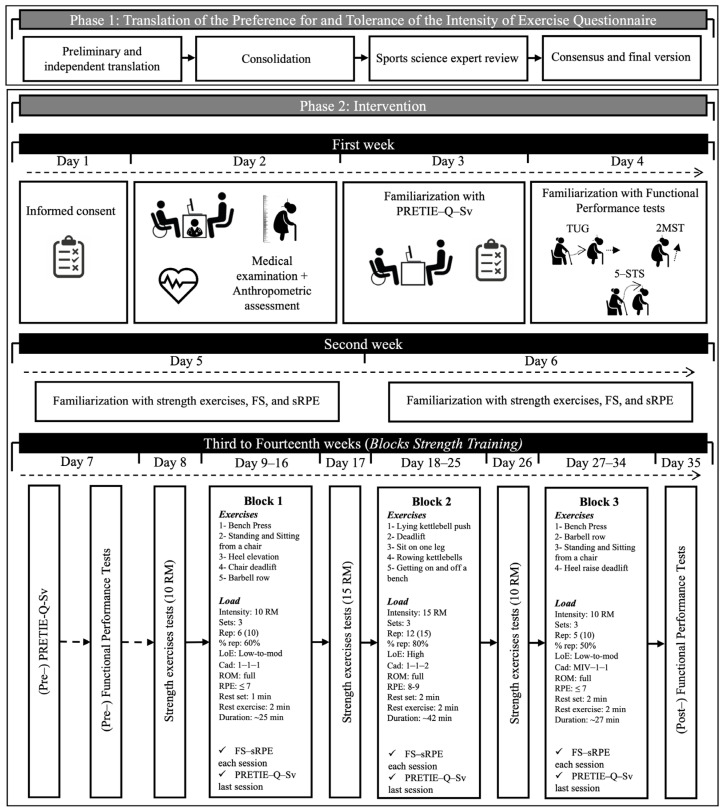
Research design. Block 1: Bilateral strength training with low to moderate level of effort. Focus: force peak; Block 2: Unilateral strength training with high level of effort. Focus: force tolerance; Block 3: Bilateral strength training with low to moderate level of effort. Focus: power peak; Cad: cadence (seconds) during concentric, isometric, and eccentric phases; FS: Feeling Scale; LoE: Level of Effort; MIV: Maximal Intended Velocity; PRETIE-Q-Sv: Preference for and Tolerance of the Intensity of Exercise Questionnaire Spanish version; Rep: repetitions; 10 RM: 10 repetition maximum; 15 RM: 15 repetition maximum; ROM: Range of Motion; RPE: Rating of Perceived Exertion; TUG: Timed Up and Go; 2MST: Two-Minutes Step Test; 5-STS: Five Times Stand-to-Sit test; %rep: percentage of the maximum number of repetitions.

**Figure 2 healthcare-13-03000-f002:**
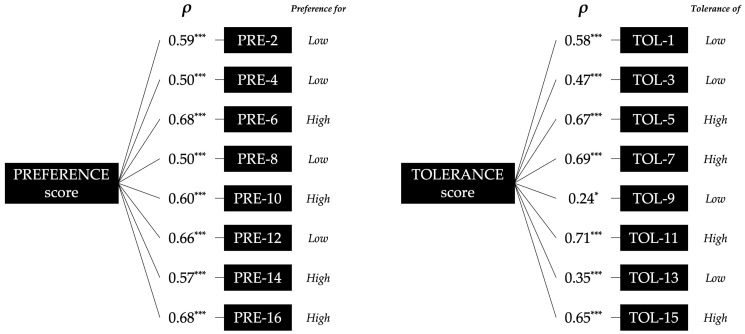
Spearman’s rank correlation coefficient (*ρ*) between individual items of the Preference and Tolerance scales and their respective total scores. The terms “low” and “high” refer to Preference for low or high exercise intensity, and to low or high Tolerance of uncomfortable intensity, respectively. * *p* < 0.05; *** *p* < 0.001.

**Figure 3 healthcare-13-03000-f003:**
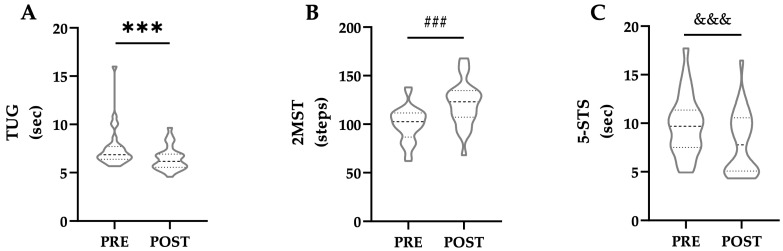
Functional performance in (**A**) Timed Up and Go (TUG), (**B**) Two-Minutes Step Test (2MST), and (**C**) Five Times Stand-to-Sit Test (5-STS) pre- and post-Block Strength Training. *** *p* < 0.001, *r_b_* = 0.94; ^###^ *p* < 0.001, *r_b_* = −1.00; ^&&&^ *p* < 0.001, *d* = 1.25.

**Table 1 healthcare-13-03000-t001:** Preference and Tolerance during the intervention.

	PRE			B1			B2			B3		
	M	±	SD	M	±	SD	M	±	SD	M	±	SD
Preference (score)	19.93	±	5.41	22.50	±	7.21	21.89	±	7.62	24.18	±	8.63
Tolerance (score)	19.93	±	4.27 ^†^	20.04	±	4.66 *	24.29	±	6.86 ^†^*	25.18	±	7.47 ^†^*

PRE: pre-intervention; B1, B2, and B3: Blocks 1, 2, 3, respectively. ^†^ *p* = 0.005 and *p* = 0.001 for PRE vs. B2 and B3, respectively. * *p* = 0.009 and *p* = 0.002 for B1 vs. B2 and B3, respectively.

**Table 2 healthcare-13-03000-t002:** Feeling Scale and Perceived Exertion during the intervention.

	B1	B2	B3
	Median (IQR)	Median (IQR)	Median (IQR)
FS (−5 to +5)	4.80 (4.60–5.00) ^#^	4.90 (4.80–5.00)	5.00 (4.90–5.00) ^#^
sRPE (0 to 10 scale)	3.50 (3.25–3.88) ^‡^	4.25 (4.13–4.41) ^‡&^	3.00 (2.63–3.13) ^‡&^

B1, B2, and B3: Blocks 1, 2, 3, respectively; FS: Feeling Scale; sRPE: session Rating of Perceived Exertion. ^#^ *p* < 0.001 for B1 vs. B3. ^‡^ *p* < 0.001 for B1 vs. B2 and B3, respectively. ^&^ *p* < 0.001 for B2 and B3.

## Data Availability

The datasets used and/or analyzed during the current study are available from the corresponding author on reasonable request.

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
