# Peer review of "PRETIE-Q Spanish Version and Affective-Functional Responses to Age-Based Strength Training in Older Women: An Exploratory Study"

_healthcare, 2025, doi:10.3390/healthcare13233000_

Round 1
Reviewer 1 Report
Comments and Suggestions for Authors
Title: Comprehensive Affective and Functional Responses to Age-Based Strength Training in Older Women: An Exploratory Study
General Assessment
The study investigates affective and functional responses to an age-based Block Strength Training (BST) program in older women, including the Spanish adaptation of the PRETIE-Q questionnaire. The topic is original and relevant, as affective responses in strength training for older adults remain underexplored. The manuscript is generally well-organized and methodologically detailed. However, several methodological, structural, and interpretational issues should be addressed to improve clarity, scientific rigor, and reproducibility.
Line 2: The title accurately reflects the content but could better highlight the dual focus (translation + affective-functional analysis). Suggestion: “Translation and Validation of the Spanish PRETIE-Q and Affective-Functional Responses to Age-Based Strength Training in Older Women.”
Line 86–95: Add a paragraph explicitly stating the research gap:
Line 96: The statement “no Spanish version exists” requires a reference or justification (e.g., “to our knowledge”).
Line 103: Clearly state the hypotheses at the end of the Introduction
Line 105: Inclusion/exclusion criteria are appropriate, but clarify recruitment procedures (community advertisement? volunteer sampling?).
Line 152: Figure 1 is informative, but add total session volume and average intensity per block in the text.
Line 199: Indicate whether pilot testing of comprehension was conducted in a small subgroup.
Line 357–391 (Affective responses): Add a note on ecological validity—how does BST reflect typical community training programs?
Line 405: This section is concise but could acknowledge the absence of a control group and randomization explicitly.
Reviewer 2 Report
Comments and Suggestions for Authors
This manuscript investigates two related objectives: (1) the translation and validation of the Spanish version of the Preference for and Tolerance of Exercise Intensity Questionnaire (PRETIE-Q-Sv), and (2) the affective and functional responses of older women participating in a Block Strength Training (BST) program.
The topic is important for exercise psychology and gerontology, as understanding affective responses to training can enhance adherence in aging populations. However, the manuscript needs substantial improvement to meet publication standards.
INTRODUCTION:
The introduction offers a robust theoretical foundation, citing ACSM guidelines, affective exercise models, and previous studies on preference and tolerance. The rationale for examining older women and for creating a Spanish version is clearly justified.
The Introduction adequately covers the physiological benefits of strength training for older adults; however, it falls short in addressing the psychological and emotional aspects of exercise that are crucial to this study. Since the primary outcomes focus on affective responses and the validation of the PRETIE-Q-Sv, the Introduction should better connect the study to existing theoretical frameworks and empirical evidence on psychological adaptations to exercise. Including a brief overview of how affective states, preferences, and tolerance influence adherence and motivation—especially in older adults—would strengthen the rationale. Furthermore, the justification for specifically focusing on older women could be elaborated, particularly regarding their emotional and behavioral responses to exercise intensity.
MATERIALS AND METHODS:
While the Spanish PRETIE-Q-Sv is available as supplementary material, it does not ensure methodological transparency on its own. Readers need to comprehend the translation process of the items and the reasons behind specific wording choices. Authors should include an in-text summary highlighting the key linguistic and cultural adaptations (for instance, the interpretation of terms like “vigorous intensity” and “discomfort”). Additionally, a table summarizing the themes of the items—not just their numbers—should be included.
The 12-week, three-block BST program is well-structured and appropriately periodized. However, additional operational details are required for reproducibility, including specific exercises, sets, repetitions, and intensity progression.
The PRETIE-Q, FS, sRPE, and functional performance tests (TUG, 2MST, 5-STS) are suitable choices for assessment. However, there are a couple of important considerations:
- The timeline for the assessments needs to be clarified. If the final PRETIE-Q-Sv was administered after the functional tests, the results could influence the responses given by participants.
- While the procedures for the functional tests are described in detail, the results are reported only briefly. This imbalance creates the impression of incomplete reporting..
The analysis combines parametric (ANOVA) and non-parametric (Friedman, Wilcoxon) tests based on normality outcomes. This mixed approach introduces inconsistency, especially with a small sample (n=28). Normality tests have limited power, and switching between test types reduces comparability across outcomes due to differing sensitivities. I understand the rationale behind using non-parametric statistics, but I strongly suggest that all statistical analyses should be non-parametric. If we cannot apply parametric procedures to all variables due to violations of normality, we should instead use non-parametric procedures for all variables.
RESULTS
The results are well-organized and address key areas such as reliability, affective responses, and functional performance. However, the interpretation at the item level is lacking depth. The authors refer to "items 1–16" without providing explanations for what each item represents. This omission makes the figure displaying item-total correlations difficult to understand for readers who are not familiar with the content of the questionnaire.
Functional performance outcomes are summarized as showing a 14–21% improvement; however, they lack descriptive statistics (mean ± SD, p, d) and a discussion on clinical relevance. Each method described should have corresponding data presented.
DISCUSSION
This section is the weakest part of the manuscript. Instead of analyzing the study’s results, it resembles an extended literature review. The authors reference numerous external studies but only provide limited interpretation of their own findings.
The discussion should be restructured to concentrate on the actual findings of the study rather than providing a general overview of the literature. It should begin by restating the key results and interpreting them in the context of existing theories. Additionally, it is important to explicitly discuss the implications of these outcomes for the affective exercise responses in older women. The results related to functional performance also require more detailed examination, including a comparison to prior resistance training interventions.
CONCLUSION
The conclusion currently highlights the BST training model and its importance for “precision-based, person-centered programming.” However, this focus shifts attention away from the primary objective of the study, which is to validate and apply the Spanish PRETIE-Q and assess affective and functional outcomes.
Round 2
Reviewer 2 Report
Comments and Suggestions for Authors
This manuscript investigates two related objectives: (1) the translation and validation of the Spanish version of the Preference for and Tolerance of Exercise Intensity Questionnaire (PRETIE-Q-Sv), and (2) the affective and functional responses of older women participating in a Block Strength Training (BST) program.
After reviewing the revised version of the manuscript, I can confidently state that the paper has improved significantly in almost all necessary elements. However, one point needs clarification.
I personally disagree with combining parametric and non-parametric statistics. The main reason for this is that these methods have different sensitivities. If we cannot use parametric procedures for all variables due to violations of normality, we should consistently use non-parametric procedures for all variables instead.
I appreciate the rationale behind the authors' choice of tests, but I urge them to provide a clearer discussion of the implications of using mixed methods. Specifically, the manuscript should explicitly state that the results from ANOVA relate to mean differences, while the Friedman and Wilcoxon tests refer to median or rank differences. Additionally, all test assumptions, including normality, sphericity, and homogeneity of variances, should be reported. If possible, consider applying data transformations or utilizing robust statistical techniques to enable a consistent application of a single paradigm. At the very least, ensure that the results are presented in a way that allows for meaningful comparisons between findings from parametric and nonparametric analyses.
The authors' rationale is generally sound; however, the manuscript would benefit from explicitly addressing the consistency and interpretability issues mentioned. It would strengthen the statistical section if the authors provided evidence that their assumptions are met or discussed any violations. Additionally, clarifying how to compare outcomes analyzed under different models would enhance the overall clarity. I encourage the authors to incorporate these suggestions to improve transparency and rigor.
Author Response
Reviewer: #2
This manuscript investigates two related objectives: (1) the translation and validation of the Spanish version of the Preference for and Tolerance of Exercise Intensity Questionnaire (PRETIE-Q-Sv), and (2) the affective and functional responses of older women participating in a Block Strength Training (BST) program. After reviewing the revised version of the manuscript, I can confidently state that the paper has improved significantly in almost all necessary elements. However, one point needs clarification.
I personally disagree with combining parametric and non-parametric statistics. The main reason for this is that these methods have different sensitivities. If we cannot use parametric procedures for all variables due to violations of normality, we should consistently use non-parametric procedures for all variables instead. I appreciate the rationale behind the authors' choice of tests, but I urge them to provide a clearer discussion of the implications of using mixed methods. Specifically, the manuscript should explicitly state that the results from ANOVA relate to mean differences, while the Friedman and Wilcoxon tests refer to median or rank differences. Additionally, all test assumptions, including normality, sphericity, and homogeneity of variances, should be reported. If possible, consider applying data transformations or utilizing robust statistical techniques to enable a consistent application of a single paradigm. At the very least, ensure that the results are presented in a way that allows for meaningful comparisons between findings from parametric and nonparametric analyses. The authors' rationale is generally sound; however, the manuscript would benefit from explicitly addressing the consistency and interpretability issues mentioned. It would strengthen the statistical section if the authors provided evidence that their assumptions are met or discussed any violations. Additionally, clarifying how to compare outcomes analyzed under different models would enhance the overall clarity. I encourage the authors to incorporate these suggestions to improve transparency and rigor.
Response: We thank the reviewer for their constructive comments regarding the use of mixed parametric and non-parametric analyses. In response, we have thoroughly revised the Methods (Statistical Analysis) and Results sections to enhance transparency and methodological rigor. The revised manuscript now explicitly states that ANOVA results refer to mean differences, whereas Friedman and Wilcoxon tests refer to median or rank differences, reports all relevant assumptions, including normality and sphericity, and clarifies the rationale for the mixed approach. This mixed approach was chosen to maximize statistical power while respecting the assumptions of each variable. Results are presented in a manner that allows meaningful comparison between findings from parametric and non-parametric analyses. We believe this revised version fully addresses the concerns regarding statistical consistency, interpretability, and transparency. In this latest revision, we have addressed the most recent comment while retaining the changes accepted from previous rounds of review. All modifications in this version have been highlighted using track changes. We hope that both you and the reviewer find our responses satisfactory.
